# *Tm*Spz6 Is Essential for Regulating the Immune Response to *Escherichia coli* and *Staphylococcus aureus* Infection in *Tenebrio molitor*

**DOI:** 10.3390/insects11020105

**Published:** 2020-02-05

**Authors:** Tariku Tesfaye Edosa, Yong Hun Jo, Maryam Keshavarz, Young Min Bae, Dong Hyun Kim, Yong Seok Lee, Yeon Soo Han

**Affiliations:** 1Department of Applied Biology, Institute of Environmentally-Friendly Agriculture (IEFA), College of Agriculture and Life Sciences, Chonnam National University, Gwangju 61186, Korea; bunchk.2000@gmail.com (T.T.E.); yhun1228@jnu.ac.kr (Y.H.J.); mariakeshavarz1990@gmail.com (M.K.); ugisaka@naver.com (Y.M.B.); kjsdh3@hotmail.com (D.H.K.); 2Ethiopian Institute of Agricultural Research, Ambo Agricultural Research Center, Ambo 37, Ethiopia; 3Department of Life Science and Biotechnology, College of Natural Sciences, Soonchunhyang University, Asan 31538, Korea; yslee@sch.ac.kr

**Keywords:** antimicrobial peptides, immune response, mealworm, spätzle, Toll pathway

## Abstract

Spätzle is an extracellular protein that activates the Toll receptor during embryogenesis and immune responses in *Drosophila*. However, the functions of the spätzle proteins in the innate immune response against bacteria or fungi in *T. molitor* are not well understood. Therefore, in this study, the open reading frame (ORF) of *TmSpz6* was identified and its function in the response to bacterial and fungal infections in *T. molitor* was investigated using RNAi. The highest expression of *TmSpz6* was in prepupae, and 3- and 6-day-old pupae, while remarkable expression was also observed in other stages. The tissue-specific expression analysis showed that *TmSpz6* expression was highest in the hemocytes of larvae. *TmSpz6* expression was highly induced when challenged with *Escherichia coli*, *Staphylococcus aureus*, or *Candida albicans* at 6 h post-injection; however, *TmSpz6-*silenced larvae were significantly more susceptible to only *E. coli* and *S. aureus* infection. The antimicrobial peptides (AMPs) gene expression analysis results show that *TmSpz6* mainly positively regulated the expression of *TmTencin-2* and *-3* in response to *E. coli* and *S. aureus* infection. Collectively, these results suggest that *Tm*Spz6 plays an important role in regulating AMP expression and increases the survival of *T. molitor* against *E. coli* and *S. aureus*.

## 1. Introduction

Insects have well-developed and complex innate immune systems that are divided into humoral and cellular components. The humoral immune system principally consists of antimicrobial peptides (AMPs), lectins, lysozyme, protease inhibitors, and other factors [1]. The recognition of invading pathogens leads to the activation of signal transduction pathways, such as Toll or immune deficiency (IMD), and ultimately induces the expression of antimicrobial peptides (AMPs) that attack the invading pathogens [2,3].

In *Drosophila*, AMP production is controlled by downstream signal transduction pathway that includes spätzle, Toll, tube, pelle, and cactus [4,5,6,7]. This pathway is activated when pattern-recognition receptors (PRRs) recognize the pathogen-associated molecular pattern [8] and convey the signal to a proteolytic cascade that leads to the cleavage of the cytokine-like protein pro-spätzle; the resulting cleaved, mature spätzle activates the Toll receptor as a ligand [9,10].

Binding of spätzle to the Toll receptor activates a downstream signaling pathway containing several protein complexes, including adaptor proteins such as myeloid differentiation primary response 88 (MyD88), Tube, and the serine-threonine innate immunity kinase (Pelle) [11]. Signaling through the MyD88–Tube–Pelle complex leads to the phosphorylation and degradation of the IκB factor Cactus, which inhibits the nuclear localization and activity of NF-κB transcription factors. Once Cactus is degraded, both Dorsal and Dif translocate to the nucleus and bind to κB-related sequences in the AMP genes [12].

Initially, spätzle exists as an inactive extracellular pro-spätzle, which contains a signal peptide, an N-terminal regulatory cleavage site, and a signaling precursor consisting of a 106-amino acid C-terminal active fragment [13,14]. The active ligand spätzle is generated during development or immune response [14].

In *Tenebrio molitor*, upon infection, the recognition of Lys-type peptidoglycan (PG) (gram-positive bacteria) or β -1,3-glucan (fungi) and DAP-type PG (gram-negative bacteria) by PGRP-SA/GNBP1 or PGRP-LE/LC activates a serine protease cascade. This cascade, which includes modular serine protease (MSP), spätzle processing enzyme activating enzyme (SAE), and spätzle-processing enzyme (SPE), mediates the cleavage of pro-spätzle and the generation of active spätzle [9,15]. The importance of spätzle in the Toll pathway has been demonstrated in insects such as *Drosophila melanogaster* [16,17], *Bombyx mori* [18], *Manduca sexta* [19], and mosquitoes [20,21], as well as shrimps, including *Fenneropenaeus chinensis* [22] and *Litopenaeus vannamei* [23].

Spätzle proteins have been shown to be important in the innate immune response to microbial infection in various arthropods from insects to shrimp. However, apart from a biochemical characterization [9], the functions of the spätzle genes in the *T. molitor* immune system in response to microbial challenge have not been yet studied. Therefore, the current study focused on the identification and functional characterization of *Tm*Spätzle 6 in innate immune response to various pathogens in *T. molitor*.

## 2. Materials and Methods

### 2.1. Insect Culture

*T. molitor* (mealworm) was maintained at 27 ± 1 °C and 60 ± 5% relative humidity in the dark on an artificial diet containing 170 g of whole-wheat flour, 20 g of fried bean powder, 10 g of soy protein, 100 g of wheat bran, 200 mL of sterile water, 0.5 g of chloramphenicol, 0.5 g of sorbic acid, and 0.5 mL of propionic acid. For the experiments, 10th–12th instar larvae were used. To ensure that uniformly sized larvae were used, the larvae were separated according to size using a set of laboratory test sieves (Pascall Eng. Co., Ltd., Crawley, Sussex, England).

### 2.2. Preparation of Microorganisms

The gram-negative bacterium *Escherichia coli* K12, the gram-positive bacterium *Staphylococcus aureus* RN4220, and the fungus *Candida albicans* were used to study the function of *TmSpz6* in the innate immune response of mealworms against microbial infection. These microorganisms have been repeatedly used by various researchers to study the immune related gene expression [24,25,26]. The microorganisms were cultured in Luria–Bertani (LB; *E. coli* and *S. aureus*) and Sabouraud dextrose (*C. albicans*) broths at 37 °C overnight and were subsequently subcultured at 37 °C for 3 h. Then, the microorganisms were harvested and washed two times by centrifugation at 3500 rpm for 10 min in phosphate-buffered saline (PBS; pH 7.0). The washed microorganisms were suspended in PBS and their optical densities were measured at 600 nm (OD_600_) to determine the concentration. Finally, 10^6^ cells/μL of *E. coli* or *S. aureus* and 5 × 10^4^ cells/μL of *C. albicans* were separately injected into the larvae.

### 2.3. Identification and Cloning of the Full-length TmSpz6 cDNA

The *T. molitor TmSpz6* gene was identified by a local-tBLASTn analysis using the amino acid sequence of the *T. castaneum spz6* gene (NP_001164082.1) as the query. The partial cDNA sequence of *Tmspz6* was obtained from the *T. molitor* RNAseq database, and the full-length cDNA sequence of *TmSpz6* was identified by the 5′- and 3′-rapid amplification of cDNA end (RACE) polymerase chain reaction (PCR) using a SMARTer RACE cDNA amplification kit (Clontech, Mountain View, CA, USA) according to the manufacturer’s instructions. The PCR was performed with AccuPower^®^ PyroHotStart Taq PCR PreMix (Bioneer, Korea) and *TmSpz6*-specific primers (*TmSpz6*-cloning_Fw and *TmSpz6*-cloning_Rv; Table 1) under the following cycling conditions: a pre-denaturation step at 95 °C for 5 min, followed by 35 cycles of denaturation at 95 °C for 30 s, annealing at 53 °C for 30 s, and extension at 72 °C for 2 min, with a final extension step at 72 °C for 5 min on a MyGenie96 Thermal Block (Bioneer, Korea). The PCR products were purified using the AccuPrep^®^ PCR Purification Kit (Bioneer, Korea), immediately ligated into T-Blunt vectors (Solgent, Korea), and transformed into *E. coli* DH5α competent cells according to the manufacturer’s instructions. Plasmid DNA was extracted from the transformed cells using the AccuPrep^®^ Nano-Plus Plasmid Extraction Kit (Bioneer, Korea), sequenced, and analyzed. Finally, the full-length cDNA sequence of *TmSpz6* was obtained.

### 2.4. Domain and Phylogenetic Analyses

Specific domains were analyzed using the InterProScan 5 and BLASTp programs. A multiple sequence alignment was performed with representative Spz6 protein sequences from other insects obtained from GenBank using ClustalX2 software. Phylogenetic and percentage identity analyses were conducted using ClustalX2 and MEGA 7 programs.

### 2.5. Analysis of TmSpz6 Expression and Induction Patterns

To study the expression of *TmSpz6* across development, total RNA was extracted from whole *T. molitor* (n = 20) eggs (EG), young instar larvae (YL; 10th–12th instar larvae), late instar larvae (LL; 19th–20th instar larvae), prepupae (PP), 1–7-day-old pupae (P1–P7), and 1–5-day-old adults (A1–A5). To investigate the tissue-specific expression patterns of *TmSpz6*, RNA was extracted from the gut, hemocytes, integument, Malpighian tubules, and fat body of late instar larvae and adults as well as the ovaries and testes of adults (n = 20). The *TmSpz6* gene expression induction pattern in *T. molitor* larval tissues was analyzed following injection with *E. coli*, *S. aureus*, or *C. albicans* into young instar larvae. Three immune response-related tissues, the hemocytes, fat body, and gut, were collected at 3, 6, 9, 12, and 24 h post-injection.

The samples were collected into 500 μL of guanidine thiocyanate RNA lysis buffer (2 mL of 0.5 M EDTA, 1 mL of 1 M MES Buffer, 17.72 g of guanidine thiocyanate, 0.58 g of sodium chloride, 0.7 mg of phenol red, 25 μL of Tween-80, 250 μL of glacial acetic acid, and 500 μL of isoamyl alcohol) and were homogenized with a homogenizer (Bertin Technologies, France) at 7500 rpm for 20 s. The total RNA was extracted using the modified LogSpin RNA isolation method [27], and 2 μg of the extracted total RNA was used to immediately synthesize cDNA with AccuPower^®^ RT PreMix (Bioneer, Korea) and Oligo (dT) 12–18 primers on a MyGenie96 Thermal Block (Bioneer, Korea) according to the manufacturer’s instructions.

Real-Time Quantitative PCR (RT-qPCR) was performed on an Exicycler^TM^ 96 Real-Time Quantitative Thermal Block (Bioneer, Daejeon, Korea) with gene-specific primers under the following cycling conditions: an initial denaturation step at 94 °C for 5 min, followed by 45 cycles of denaturation at 95 °C for 15 s and annealing at 60 °C for 30 s. The 2^−ΔΔCt^ method [28] was used to analyze *TmSpz6* expression levels. *T. molitor* ribosomal protein L27a (*TmL27a*) was used as an internal control to normalize differences in template concentration between samples.

### 2.6. RNA Interference Analysis

To synthesize the double-stranded (ds) RNA of the *TmSpz6* gene, primers containing the T7 promoter sequence at their 5′ ends were designed using SnapDragon-Long dsRNA design software (Table 1). The PCR was conducted using AccuPower^®^ Pfu PCR PreMix with the TmSpz6_Fw and TmSpz6_Rv primers (Table 1) under the following cycling conditions: an initial denaturation step at 94 °C for 2 min followed by 35 cycles of denaturation at 94 °C for 30 s, annealing at 53 °C for 30 s, and extension at 72 °C for 30 s, with a final extension step at 72 °C for 5 min. PCR products were purified using the AccuPrep PCR Purification Kit (Bioneer, Daejeon, Korea), and dsRNA was synthesized using the Ampliscribe^TM^ T7-Flash^TM^ Transcription Kit (Epicentre Biotechnologies, Madison, WI, USA) according to the manufacturer’s instructions. After synthesis, the dsRNA was purified by precipitation with 5 M ammonium acetate and 80% ethanol, and then it was quantified with an Epoch spectrophotometer (BioTek Instruments, Inc., Winooski, VT, USA). The dsRNA for enhanced green fluorescent protein (ds*EGFP*) was synthesized for use as a control and was stored at −20 °C until use.

### 2.7. Effect of TmSpz6 Gene Silencing on the Response to Microorganism Challenge

To study the importance of TmSpz6 in the *T. molitor* immune response, ds*TmSpz6* (1 µg/µL) was first injected into young-instar larvae (10th–12th instars; n  =  30) using disposable needles mounted onto a micro-applicator (Picospiritzer III Micro Dispense System; Parker Hannifin, Hollis, NH, USA). An equal amount of ds*EGFP* was injected in similar larvae as a negative control. *TmSpz6* knockdown was evaluated, and over 90% knockdown was achieved at 2 days post-injection. Second, the *TmSpz6*-silenced and ds*EGFP*-injected larval groups were challenged with *E. coli* (10^6^ cells/μL), *S. aureus* (10^6^ cells/μL), or *C. albicans* (5 × 10^4^ cells/μL) in triplicate experiments. The challenged larvae were maintained for 10 days, during which the number of surviving larvae were recorded. The survival rates of the *TmSpz6*-silenced larvae were compared to survival of control larvae.

### 2.8. Effect of dsTmSpz6 on AMP Expression in Response to Microbial Challenge

To characterize the function of TmSpz6 in regulating AMP gene expression in response to microbial infection, *TmSpz6* gene expression in larvae was silenced with RNAi and then these larvae were injected with microorganisms (*E. coli*, *S. aureus*, or *C. albicans*). ds*EGFP* and PBS were used as the negative and injection controls, respectively. At 24 h post-injection, the hemocytes, fat body, and gut were dissected, and then total RNA was extracted from these tissues and cDNA was synthesized as described above. Next, qRT-PCR was conducted using specific primers (Table 1) to analyze the temporal expression patterns of 14 AMP genes: *TmTen*cin-1 (*TmTen-1*), *TmTen*cin-2 (*TmTen2*), *TmTen*cin-3 (*TmTen-3*), *TmTen*cin-4 (*TmTen-4*), *TmAttacin-1a* (*TmAtt-1a*), *TmAttacin-1b* (*TmAtt-1b*), *TmAttacin-2* (*TmAtt-2*), *TmDefensin-1* (*TmDef-1*), *TmDefensin-2* (*TmDef-2*), *TmColeoptericin-1* (*TmCol-1*), *TmColeoptericin-2* (*TmCol-2*), *TmCecropin-2* (*TmCec-2*), *TmThaumatin like protein-1* (*TmTLP-1*), and *TmThaumatin like protein-2* (*TmTLP-2*).

### 2.9. Data Analysis

Statistical analyses were conducted using SAS 9.4 software (SAS Institute, Inc., Cary, NC, USA), and cumulative survival was analyzed by Tukey’s multiple test, at a significance level of *p* < 0.05. AMP gene expression was calculated using the delta Ct (ΔΔCt) method, and the fold change compared to the internal control (*TmL27a*) and external control (PBS) was calculated by the 2 ^(ΔΔCt)^ method.

## 3. Results

### 3.1. Sequence Identification and Phylogenetic Analysis of TmSpz6

The full-length open reading frame (ORF) sequence of *TmSpz6* was obtained from the *T. molitor* RNAseq database and by 5′- and 3′-RACE PCR. The *TmSp6* ORF is 1227 bp and encodes 408 amino acid residues (Figure 1). Domain analysis indicated that *Tm*Spz6 contains one cystine-knot domain at the C-terminus, one cleavage site, and one signal peptide region. Phylogenetic analysis revealed that *Tm*Spz6 grouped together with another spätzle 6 from a Coleopteran insect (*Tribolium castaneum* spätzle 6) (Figure 2).

### 3.2. Developmental and Tissue-Specific Expression Patterns of TmSpz6

The developmental and tissue-specific expression patterns of *TmSpz6* mRNA were examined by RT-qPCR. *TmSpz6* transcripts were observed in all tested developmental stages and tissues (Figure 3). *TmSpz6* was expressed at similar levels in young and late larvae, but was highly expressed in pre-pupae. Expression was decreased in 1- and 2-day-old pupae but was then highly upregulated and peaked in 3-day-old pupae. This was followed by a gradual decrease in expression in 4- and 5-day-old pupae and an increase in 6-day-old pupae. During the adult stage, *TmSpz6* was expressed at similar levels at all examined ages, except it was slightly lower in 5-day-old adults (Figure 3A).

The examination of different tissues revealed that *TmSpz6* was highly expressed in hemocytes, but was expressed at lower levels in the integument, fat body, and Malpighian tubules of late instar larvae (Figure 3B). In adults, expression was high in the integument and fat body followed by the testis and hemocytes (Figure 3C). Comparatively low *TmSpz6* expression levels were detected in the gut, Malpighian tubules, and ovaries of 5-day-old adults.

### 3.3. Temporal Induction of TmSpz6 after Microbial Challenge

To determine the inducibility of *TmSpz6* during microbial infection, the temporal expression of *TmSpz6* in *T. molitor* larvae was examined after injecting *E. coli*, *S. aureus*, or *C. albicans*. The temporal expression patterns in three immune-related tissues, hemocytes, fat body, and gut, were analyzed at 3, 6, 9, 12, and 24 h post-injection by RT-qPCR. The microbial challenge time-dependently induced transcription of *TmSpz6* in all tested tissues. All injected microorganisms induced the highest *TmSpz6* expression in hemocytes at 6-h post injection (Figure 4A). In *E. coli-* and *S. aureus-*injected larvae, the highest *TmSpz6* expression in the gut was observed at 9 h post injection (Figure 4B), whereas in *E. coli-* and *C. albicans-*injected larvae, the highest expression in the fat body was detected at 24 h post-injection (Figure 4C).

### 3.4. Effect of TmSpz6 Silencing on T. molitor Survival

Since the temporal induction of *TmSpz6* was observed following microorganism injection, the survival of *TmSpz6-*silenced *T. molitor* larvae in response to infection was assessed. The survival of *TmSpz6*-silenced or ds*EGFP-*injected *T. molitor* larvae after microbial injection was monitored for 10 days. The results show that the injection of ds*TmSpz6* or d*EGFP* did not affect the survival of *T. molitor* larvae after injection with PBS. However, ds*TmSpz6-*injected larvae were highly susceptible to *E. coli* (68.5%) and *S. aureus* (57.5%) (Figure 5B,C), whereas their survival rate after infection with *C. albicans* was not significantly different from that of the controls (Figure 5D).

### 3.5. Effects of TmSpz6 Gene Silencing on the Expression of AMP Genes

Based on the results of the survival study, the importance of *TmSpz6* in the immune defense against gram-negative and gram-positive bacteria was postulated. Thus, the function of *TmSpz6* in the production of AMPs in response to microbial infection was investigated by silencing the expression of *TmSpz6* in *T. molitor* larvae, challenging them with *E. coli*, *S. aureus*, or *C. albicans*, and assessing the expression levels of 14 different AMP genes at 24 h post-infection.

The results showed that *TmSpz6* silencing significantly downregulated mainly the expression of *TmTen-2* in hemocytes (Figure 6A) and gut (Figure 6B). In addition, the expression of *TmTen-3* in fat body was significantly downregulated (Figure 6C). Similarly, *TmCec-2* in hemocytes (Appendix A), *Tene-4*, *TmTLP-1*, and *TmCec-2* in the fat body (Appendix A); and *TmTen-1*, *-2*, *-4*, *TmCol-1*, *TmTLP-1*, *TmTLP-2*, and *TmCec-2* in the gut (Appendix A) were significantly downregulated following the injection with *E. coli*. Similarly, *TmCec-2* in hemocytes (Appendix A); *TmTen-2*, *-3*, *TmAtt-1a*, *TmAtt-1b*, *TmDef-1*, *TmDef-2*, *TmTLP-2*, and *TmCec-2* in the fat body (Appendix A); and *TmTen-1*,-*2*,-*3*,-*4*, *TmAtt-1a*, *TmAtt-1b*, *TmCol-1*, *TmCol-2*, *TmDef-2*, and *TmCec-2* in the gut (Appendix A) were considerably suppressed in *TmSpz6-*silenced larvae injected with *S. aureus*.

Following *C. albicans* injection, *TmTen-1*, *-4*, *TmAtt-1a*, *-1b*, *-2*, *TmCol-1*, *TmDef-1*, *-2*, and *TmCol-2* were significantly downregulated in the hemocytes of *TmSpz6*-silenced larvae (Appendix A). Additionally, *TmDef-2*, *TmTLP-2*, and *TmCec-2* in the fat body (Appendix A), and *TmTLP-1*, and *TmTLP-2* in the gut (Appendix A) were markedly downregulated in *TmSpz6*-silenced larvae. Interestingly, in contrast, *TmSpz6* knockdown increased the mRNA levels of *TmTen-1*, *-3*, *TmAtt-1a*, *-1b*, *-2*, *TmCol-1*, *-2*, and *TmDef-1* in the hemocytes of *E. coli-*challenged larvae.

## 4. Discussion

AMPs are essential effectors of the innate immune defense system against numerous pathogens [29,30]. In insects, the expression of AMPs is mainly regulated by the evolutionarily conserved Toll and IMD pathways [12]. Specifically, the Toll pathway, a well-known activator of AMP production in insects, is activated by the extracellular ligand spätzle [9]. The functions of the spätzle proteins in the innate immunity of various insect species have been repeatedly reported. In *Drosophila*, the most widely studied insect species, the functions of spätzle proteins in the defense response against microbial infections have been well studied [5,17,31]. The importance of spätzle in the activation of the Toll receptor and the subsequent activation of AMP production has also been studied in the lepidopteran insects, *B. mori* [32], and *M. sexta* [33].

In this study, *TmSpz6* was identified in *T. molitor* and functionally characterized. *TmSpz6* mRNA was highly expressed in the prepupal and pupal stages, whereas comparatively lower expression levels were observed in the larval and adult stages. Tissue-specific expression studies showed that *TmSpz6* was expressed at various levels in all examined tissues in the larval and adult stages. The observed variations in *TmSpz6* expression in different tissues and developmental stages is assumed to be controlled by developmental hormones. This assumption is based on previous studies demonstrating the importance of juvenile hormone (JH) and ecdysone in the insect innate immune response. For example, ecdysone was shown to activate *PGRP-LC* expression and induce AMP expression in *Drosophila* [34], JH was shown to act as an AMP-activator in *B. mori* [35], and ecdysone enhanced the *B. mori* immune system [36]. Therefore, it is possible that there are similar hormonal effects on *TmSpz6* expression in *T. molitor*, caused by either ecdysone or other developmental hormones. Thus, the fluctuations in the expression of *TmSpz6* at the same developmental stage or different stage are assumed to be due to hormonal changes. Briefly, the expression of *TmSpz6* was highest at the perpupal stage and in 3- and 6-day-old pupae. These are transitional stages that the ecdysone commonly controls. The investigation of the expression of ecdysone during the development of *Drosophila* by radioimmune assay showed that ecdysone activity was highest during the pupal, prepupal, and late larval stages (in descending order) [37] In addition, *TmSpz6* expression was highest in the hemocytes of larvae and adults. Previously, it has been reported that hemocytes play important roles in immunity, nutrient transportation, and growth hormone synthesis [38,39]. Furthermore, hemocytes are known to express spätzle proteins [40] and our results confirm this since we found the highest levels of *TmSpz6* expression in hemocytes, suggesting that this tissue is important in *T. molitor* immune response. Taken together, these data suggest that *TmSpz6* expression during transitional stages and in growth hormone-synthesizing tissues (hemocytes) under normal conditions is regulated by developmental hormones.

The active form of spätzle is generated during development [41] and/or the immune response to microbial infection [12,42]. Specifically, PGRP-SA or GNBP1 recognizes conserved bacterial or fungal molecules and activates the serine protease cascade that leads to the generation of mature spätzle [8,9]. The Toll receptor for the spätzle ligand is known to be located on the plasma membrane of fat body and hemocyte cells [43,44,45]. Therefore, to examine the induction of the *TmSpz6*-related immune response in *T. molitor*, larvae were challenged with *E. coli*, *S. aureus*, or *C. albicans* and then the expression of *TmSpz6* was assessed in the main immune organs, such as hemocytes, fat body, and the gut. *TmSpz6* was found to be highly induced at different time points following the injection of all tested microorganisms. Spätzle genes are activated upon microorganism infection as an effector of the Toll pathway. In *M. sexta*, spätzle-1 was upregulated at 24 h post-injection of *E. coli* or insoluble β-1,3-glucan [33]. In the current study, the highest *TmSpz6* expression was observed in hemocytes against the three pathogens. The well-known gram-negative bacteria, *E. coli* possesses DAP-type PGN, which is recognized by PGRP-LC or -LE to activate the IMD pathway in *Drosophila* [46]. However, it is not surprising that a gram-negative bacterium (*E. coli*) induced the expression of *TmSpz6* in current study, as it has already been reported that *Tenebrio* PGRP-SA can recognize the polymeric DAP-type peptidoglycan present in *E. coli* to activate the Toll receptor ligand spätzle in *T. molitor* [47].

The importance of *TmSpz6* in the immune response against microbial infection was studied using RNAi. *TmSpz6*-silenced larvae showed significant susceptibility starting from 4th day of *E. coli* injection. This indicates that *Tm*Spz6 plays a key role in the microbial defense response in *T. molitor* larvae. The expression of insect AMPs is mainly regulated by the evolutionarily conserved Toll and IMD pathways. Particularly, the spätzle–Toll–MyD88–Tube–Pell complexes and intracellular Dorsal and Dif binding to κB-related sequences play key roles in AMP production [12]. Spätzle plays a paramount role in activating Toll and is itself activated by other complexes [9]. Additionally, AMP gene expression was examined in *TmSpz6*-silenced larvae to investigate its function in the expression of AMP genes in response to microbial infection. The results reveal that *TmSpz6* silencing suppressed the induction of several AMP gene expressions in response to microbial infection, supporting the survival study results. Most of the AMP genes that were suppressed when *TmSpz6* was silenced encode AMPs that have activity against gram-negative bacteria, such as *E. coli* (*Tene-2*, *Cec-2*, *Tene-4*, and *Cole-1*) [24,47,48,49,50], and gram-positive bacteria, such as *S. aureus* (*Tene-3*, *Def-1*, *Def-2*, *Cec-2* and *Tene-1*) [51,52,53,54].

## 5. Conclusions

This study focused on the identification and immunological functional characterization of *TmSpz6* in *T. molitor*. *TmSpz6* is highly expressed in prepupal and 3-day-old pupal stages. In the tissue specific expression study, *TmSpz6* was highly expressed in the hemocytes of late larvae. The induction patterns in response to microbial challenges revealed that *TmSpz6* was highly upregulated in hemocytes at the 6 h-post injection of *E. coli*, *S. aureus*, and *C. albicans.* The survivability of *TmSpz6*-silenced larvae significantly decreased after the injection of *E. coli* or *S. aureus*. The loss of function of TmSpz6 significantly reduced the induction of various AMPs, particularly *TmTen*-2 and -3 in response to *E. coli* and *S. aureus* infection. Taken together, these results suggest that *Tm*Spz6 plays an important role in regulating AMP expression and increases the survival of *T. molitor* against *E. coli* and *S. aureus*.

## Figures and Tables

**Figure 1 insects-11-00105-f001:**
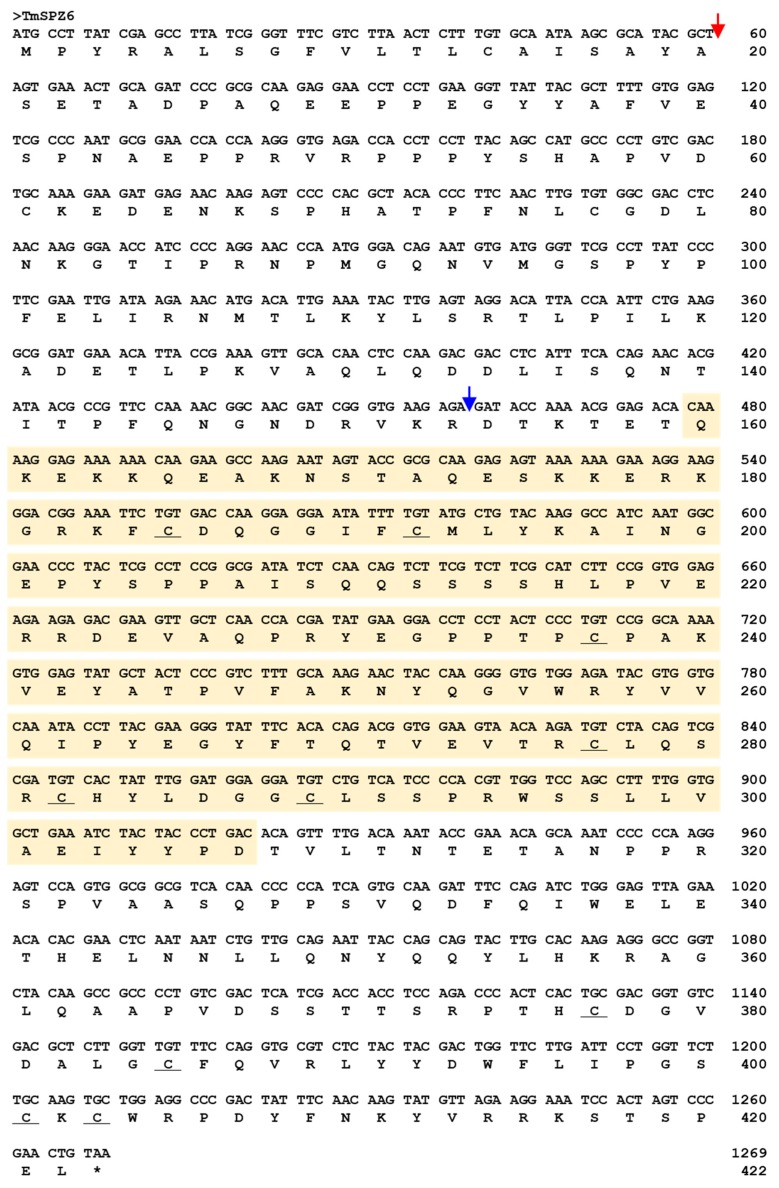
Nucleotide and deduced amino acid sequences of *TmSpz6.* It consists of a 1227 bp open reading frame encoding a 408 amino acid residue protein. Domain analysis showed that *Tm*Spz6 includes one cystine-knot domain (yellow box), one signal peptide (red arrow), and one cleavage site (blue arrow).

**Figure 2 insects-11-00105-f002:**
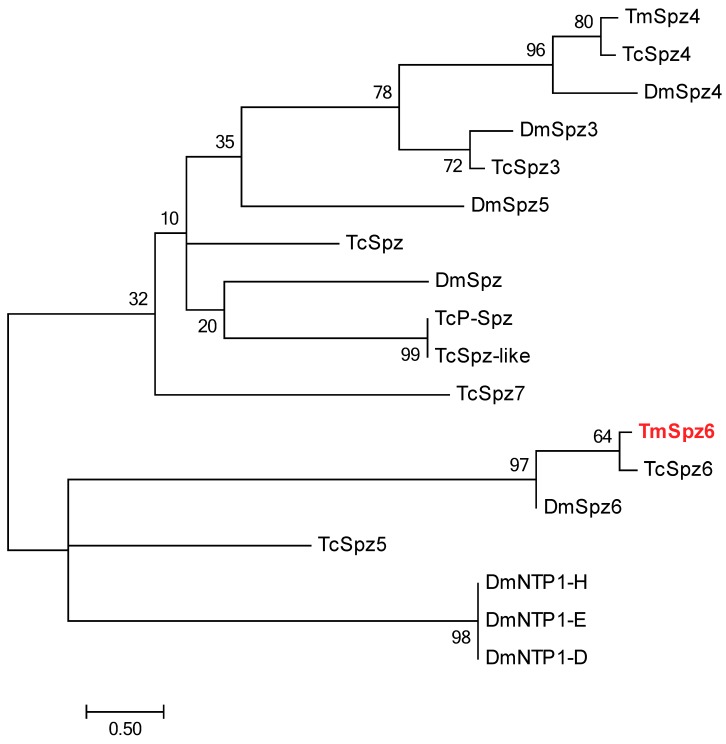
Molecular phylogenetic analysis of insect Spz6 homologs. Phylogenetic analyses of *Tm*Spz6 homologues were performed using ClustalX2, and the phylogenic tree was constructed with MEGA7 using the maximum likelihood method and 1000 bootstrapped replications. The following protein sequences were used to construct the phylogenetic tree. *Dm*Spz4, *Drosophila melanogaster* spätzle 4 (AAF53100.2); *Dm*Spz6, *D. melanogaster* spätzle 6 (AAF47261.1); *Dm*Spz5, *D. melanogaster* spätzle 5 (AAF47694.1); *Dm*Spz, *D. melanogaster* spätzle (AAF82745.1); *Dm*Spz3, *D. melanogaster* spätzle 3 (AAF52574.2); *Dm*NTP1-H, *D. melanogaster* neurotrophin 1, isoform H (AGB94113.1); *Dm*NTP1-E, *D. melanogaster* neurotrophin 1, isoform E (ACZ94621.1); *Dm*NTP1-D, *D. melanogaster* neurotrophin 1, isoform D (NP_001163348.1); *Tc*Spz7, *Tribolium castaneum* spätzle 7 (EEZ99267.2); *Tc*Spz4, *T. castaneum* spätzle 4 (EFA09263.2); *Tc*Spz5, *T. castaneum* spätzle 5 (EEZ97725.1); *Tc*Spz3, *T. castaneum* spätzle 3 (NP_001153625.1); *Tc*Spz6, *T. castaneum* spätzle 6 precursor (NP_001164082.1); *Tc*Spz, *T. castaneum* spätzle (EEZ99207.1); *Tc*P-Spz, *T. castaneum* PREDICTED protein spätzle (XP_008201191.1); *Tc*Spz-like, *T. castaneum* spätzle-like protein (EEZ99268.2).

**Figure 3 insects-11-00105-f003:**
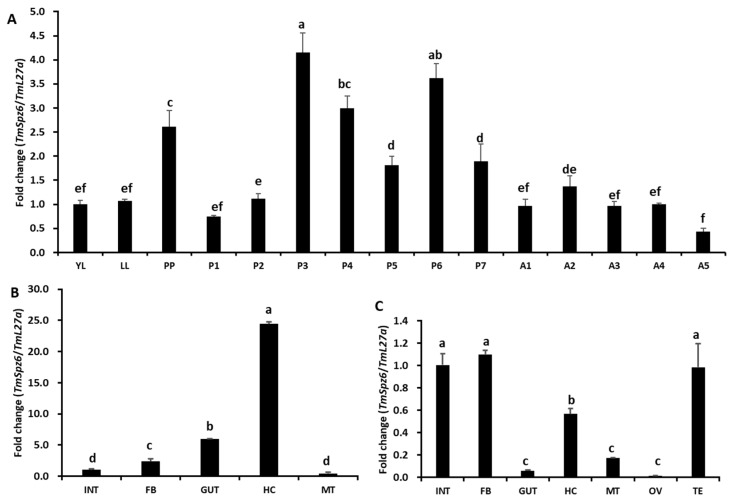
Developmental stage- and tissue-specific expression patterns of the *TmSpz6* gene in *T. molitor. TmSpz6* expression patterns across the developmental stages of *T. molitor*, including, young larvae (YL), late larvae (LL), pre-pupa (PP), 1–7-day-old pupae (P1–7), and 1–5-day-old adults (A1–5) were examined (**A**). For each sample, RNA extracted from 20 individuals was used to synthesis cDNA. The results indicate that *TmSpz6* expression gradually increased from the young larval to pre-pupal stage. In the pupal stage, the highest expression was observed in 4-day-old pupae. In adults, there was no considerable difference in expression in 2–5-day-old adults. The tissue-specific expression patterns of *TmSpz6* were also assessed in late larvae (**B**) and 5-day-old adults (**C**). The hemocytes (HC), gut, fat body (FB), Malpighian tubules (MT), and integument (INT) (for late instar larvae and adults), as well as the testes (TE) and ovaries (OV) (for adults), were dissected and collected from 20 late larvae and 5-day-old adults. *T. molitor* 60S ribosomal protein L27a (*TmL27a*) served as an endogenous control to normalize RNA levels between samples. The data are the means of three biological replicates. The asterisks indicate significant differences (*p* ≤ 0.05).

**Figure 4 insects-11-00105-f004:**
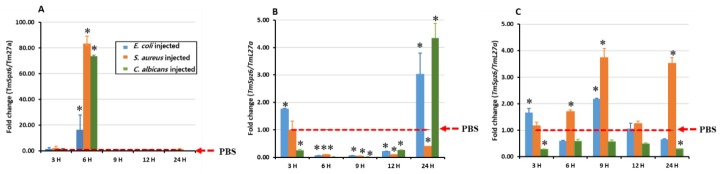
Induction patterns of *TmSpz6* expression in different *T. molitor* larval tissues. The temporal expression of *TmSpz6* was analyzed in the hemocytes (**A**), gut (**B**), and fat body (**C**) of young larvae at 3, 6, 9, 12, and 24 h post-injection of *E. coli* (10^6^ cells/μL), *S. aureus* (10^6^ cells/μL), and *C. albicans* (5 × 10^4^ cells/μL). Twenty young larvae were used at each time point. *TmSpz6* expression levels were normalized to those in 1X phosphate-buffered saline (1X PBS)-injected controls. *T. molitor* 60S ribosomal protein L27a (*TmL27a*) was used as an internal control. The asterisks indicate significant differences (*p* ≤ 0.05).

**Figure 5 insects-11-00105-f005:**
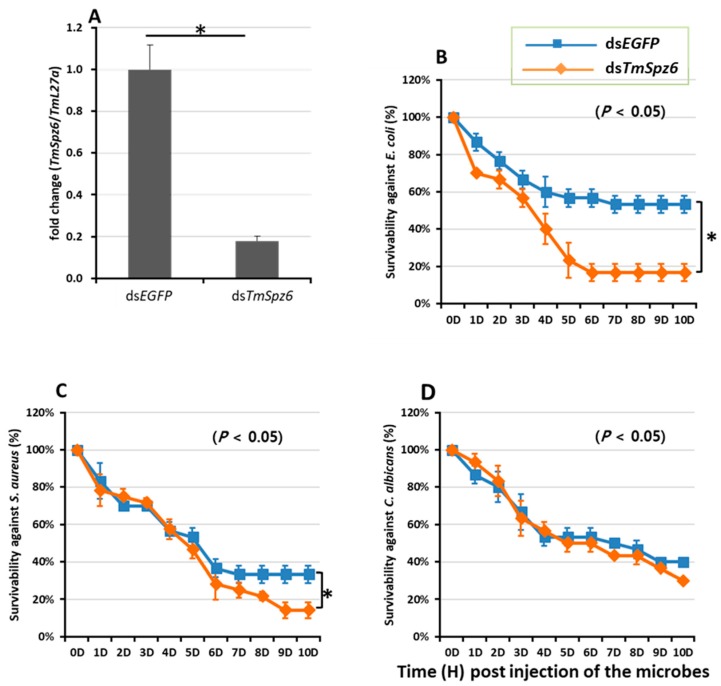
Effect of ds*TmSpz6* on the survival of *T. molitor* larvae. Silencing efficiency of *TmSpz6* mRNA was measured by qRT-PCR at 3 days post-injection (**A**). Then, the *TmSpz6*-silenced larvae were injected with *E. coli* (**B**), *S. aureus* (**C**), and *C. albicans* (**D**). ds*EGFP*-injected larvae were used as a negative control. The data are an average of three independent biological replicate experiments. The asterisks indicate significant differences between ds*TmSpz6-* and ds*EGFP-*treated groups (*p* < 0.05).

**Figure 6 insects-11-00105-f006:**
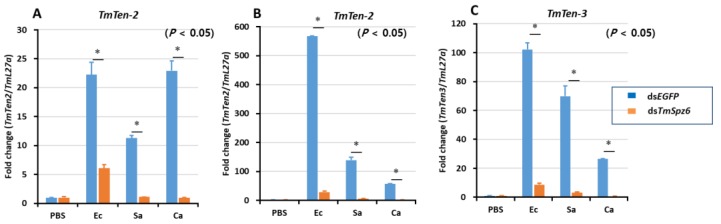
The antimicrobial peptide expression levels in *TmSpz6*-knockdown *T. molitor* larval were performed by injecting either *E*. *coli* (Ec), *S. aureus* (Sa), or *C. albicans* (Ca) on the 3th day post-*TmSpz6* silencing. 24 h post microbial challenge, the known immune tissues including hemocytes (**A**), gut (**B**), and fat body (**C**) were dissected. Then the expression levels of AMPs including *TmTenecin-1 (TmTene-1), TmTenecin-2(TmTene-2), TmTenecin-3 (TmTene-3), TmTenecin-4 (TmTene-4), TmAttacin-1a (TmAtt-1a), TmAttacin-1b (TmAtt-1b), TmAttacin-2 (TmAtt-2), TmDefensin-1 (TmDef-1), TmDefensin-2 (TmDef-2), TmColeptericin-1 (TmCole-1), TmColeptericin-2 (TmCole-2), TmCecropin-2 (TmCec-2), TmTLP-1 (TmTLP-1)* and *TmTLP2 (TmTLP-2)* were measured by qRT-PCR. dsEGFP was used as a negative control and *TmL27a* was used as an internal control. All experiments were triplicated. The asterisks represent significant differences between ds*TmSpz6-* and ds*EGFP-*treated groups when compared by the Student’s *t*-test (*p* ≤ 0.05).

**Table 1 insects-11-00105-t001:** Primers used in this study.

Primer Name	Sequence (5′-3′)
TmSpz6-qPCR-Fw	GCACAACTCCAAGACGACCT
TmSpz6-qPCR-Rv	TCTCTTCACCCGATCGTTGC
TmSpz6-T7-Fw	TAATACGACTCACTATAGGGTACCGCGCAAGAGAGTAAAAA
TmSpz6-T7-Rv	TAATACGACTCACTATAGGGTACGTATCTCCACACCCCTTG
TmSpz6-cloning-Fw	CCCCTGTCGACTGCAAAGAA
TmSpz6-cloning-Rv	CACCACGTATCTCCACACCC
TmSpz6-cloning-FullORF-Fw	TGAGTGAATAATTTCGAAAAGAAAAA
TmSpz6-cloning-FullORF-Rv	TGGGCGTTCAGTTACATCAA
TmL27a_qPCR_Fw	TCATCCTGAAGGCAAAGCTCCAGT-3′
TmL27a_qPCR_Rv	AGGTTGGTTAGGCAGGCACCTTTA-3′
dsEGFP_Fw	TAATACGACTCACTATAGGGTCGTAAACGGCCACAAGTTC
dsEGFP_Rv	TAATACGACTCACTATAGGGTTGCTCAGGTAGTGTTGTCG
TmTencin-1_Fw	CAGCTGAAGAAATCGAACAAGG
TmTencin-1_Rv	CAGACCCTCTTTCCGTTACAGT
TmTencin-2_Fw	CAGCAAAACGGAGGATGGTC
TmTencin-2_Rv	CGTTGAAATCGTGATCTTGTCC
TmTencin-3_Fw	GATTTGCTTGATTCTGGTGGTC
TmTencin-3_Rv	CTGATGGCCTCCTAAATGTCC
TmTencin-4_Fw	GGACATTGAAGATCCAGGAAAG
TmTencin-4_Rv	CGGTGTTCCTTATGTAGAGCTG
TmDefensin-1_Fw	AAATCGAACAAGGCCAACAC
TmDefencin-1_Rv	GCAAATGCAGACCCTCTTTC
TmDefencin-2_Fw	GGGATGCCTCATGAAGATGTAG
TmDefencin-2_Rv	CCAATGCAAACACATTCGTC
TmColoptericin-1_Fw	GGACAGAATGGTGGATGGTC
TmColoptericin-1_Rv	CTCCAACATTCCAGGTAGGC-3
TmColoptericin-2_Fw	GGACGGTTCTGATCTTCTTGAT
TmColoptericin-2_Rv	CAGCTGTTTGTTTGTTCTCGTC
TmAttacin-1a_Fw	GAAACGAAATGGAAGGTGGA
TmAttacin-1a_Rv	TGCTTCGGCAGACAATACAG
TmAttacin-1b_Fw	GAGCTGTGAATGCAGGACAA
TmAttacin-1b_Rv	CCCTCTGATGAAACCTCCAA
TmAttacin-2_Fw	AACTGGGATATTCGCACGTC
TmAttacin-2_Rv	CCCTCCGAAATGTCTGTTGT-3
TmCecropin-2_Fw	TACTAGCAGCGCCAAAACCT
TmCecropin-2_Rv	CTGGAACATTAGGCGGAGAA
TmThaumatin-like protein-1_Fw	CTCAAAGGACACGCAGGACT
TmThaumatin-like protein-1_Rv	ACTTTGAGCTTCTCGGGACA
TmThaumatin-like protein-2_Fw	CCGTCTGGCTAGGAGTTCTG
TmThaumatin-like protein-2_Rv	ACTCCTCCAGCTCCGTTACA
Underline indicates T7 promoter sequences

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
