# Peer review of "TmSpz6 Is Essential for Regulating the Immune Response to Escherichia coli and Staphylococcus aureus Infection in Tenebrio molitor"

_insects, 2020, doi:10.3390/insects11020105_

Round 1
Reviewer 1 Report
To the Authors
After a carefully reading of the manuscript, I concluded that it is appropriated for publication, but after some corrections. My main concern referred to the Discussion and Conclusions. In the Discussion it would be very interesting expanding the considerations about the expression of spatzle during development. It seems very likely that this gene is hormonally controlled, as suggested Figure 3. Another concern, the conclusions are also very poor. I have few other comments that you can find below.
Abstract
Line 24-26 – “In addition, AMP gene expression … and S. aureus infection”.
Comment: Please rewrite this phrase, it is not clear enough to understand the meaning.
Line 32: … well-developed innate immune …
Change to: … well-developed and complex innate immune …
Line 39-40: “… gram-negative bacteria binding … cascade leads …”
Concern: Please confirm the information related to reference 8, and if necessary, correct it.
Line 146-155: The Authors did not mention whether injection of dsGFP had any effect on the larvae.
Line 219-220 : It was discussed a possible hormonal control during development/metamorphosis?
Line 309-314: It seems clear from the results that spaetzle is hormonally controlled. The authors could have discussed it more widely, looking more carefully the results shown in Figure 3.
Line 325-328: At this point, the discussion is not clear.
Line 345: … highly expressed in perpupal ..
Please correct: … highly expressed in prepupal …
Conclusions:
The Conclusions item is relatively poor, compared to the presented results.
Reviewer 2 Report
In this study, Edosa and colleagues investigate TmSpz6 function in the response to bacterial and fungal infections in T. molitor using RNAi. The study was well designed and only minor edits are recommended prior to publication.
Minor Comments:
Abstract: ORF and AMP should be spelled out before their short-hand is used.
(Line 209-213): In the discussion, it might be informative if the authors speculate as to why they see localized expression patterns of TmSpz6. Perhaps regions where lower expression is observed represent areas where a higher abundance of native / communal bacterial populations reside?
Figure 4: The figure panel designations (A, B, C) need to be moved further away from the actual figures, preferably to the left of the Y-axis on reach graph, so they are more easily discernible.
Figure 4: Given the large difference between TmSpz6 induction from S. aureus as compared to E. coli in the hemocytes, the authors might wish to comment on the difference in peptidoglycan abundance between gram positive and negative organisms. Similarly, do the authors think that the differences in induction for a gram positive vs gram negative microbes in the gut / fat can be explained by tissue-specific abundances of immune receptors that differentiate between gram positive and gram negative bacteria? Does the microbiome of the T. molitor gut favor gram positives / gram negatives?
Figure 5: The authors should consider listing the first time point in each graph where they see a significant difference between dsEGFP and dsTmSpz6. The effect of dsTMSpz6 appears to be more dramatic and immediate for E. coli and this would appear to fit well with what is observed in the gut expression data (Figure 4b). As presented, the data appears to indicate that TmSpz6 is more important for protection against Gram negative pathogens, however, the kill kinetics of the 2 bacterial species are different, making accurate comparisons challenging.
Figure 6: E. coli infection appears to result in the maximum expression levels as compared to S. aureus. Is it known whether TmTen-2 is more effective at targeting Gram positive or Gram negative bacteria?
E. coli and S. aureus have very different doubling times in vivo. Have the authors considered this when interpreting their results? Why were the bacterial #s used chosen for the subsequent survival / gene expression experimentation (a justification should be added to the methodology section)? Have they considered utilizing heat-killed cultures or peptidoglycan purified from both E. coli and S. aureus for use in these studies to ensure that bacterial replication differences do not significantly interfere with their results?
